# How did the urban and rural resident basic medical insurance integration affect medical costs?—Evidence from China

Chen Liu[1,2], Qun Su[2]*, Meng Wang[2], Huaizhen Xing[2]

1 College of Business, Xuzhou University of Technology, Xuzhou, Jiangsu, China, 2 College of Economics and Management, Nanjing Agricultural University, Nanjing, Jiangsu, China

* suqun_njau@sina.com

## Abstract

The Urban and Rural Residents' Basic Medical Insurance (URRBMI) aims to improve access to medical services, increase medical insurance benefits to reduce medical costs, and ultimately achieve medical equity. However, in the practice of the policy, the medical costs of Chinese residents have not been reduced. To assess the impact of URRBMI on Chinese residents' healthcare resource utilization and medical costs, this study explores the fixed-effects DID methodology using CHARLS data for 2013, 2015, 2018, and 2020, explains the reasons for the rise in healthcare costs in terms of two paths, namely, demand release and moral hazard. The empirical results show that: (1) URRBMI integration increases outpatient OOP costs, inpatient OOP costs, and total medical consumption, but it also releases the population's demand for medical care, which is meaningful at the initial stage of policy implementation. (2) The sources of the increase in medical costs are the release of medical demand and moral hazard. Specifically, for the low- and middle-income groups, the integration of URRBMI triggers an increase in medical costs mainly through the release of demand. For insured persons in the high-income group, URRBMI integration increases medical costs by triggering moral hazard issues. (3) URBMI integration changes the medical resource utilization behavior of the population, prompting the population to utilize higher-level medical resources more than primary care resources. (4) In addition, integration has a positive impact on residents' preventive medical care.

## Introduction

At present, the Chinese healthcare system is confronted with the challenge of rapidly escalating medical costs. Total healthcare expenditure and health insurance spending in China are experiencing rapid growth [1]. From 2011 to 2021, the annual average growth rate of total healthcare expenditure in China was 12.2% [2]. Simultaneously, per capita, overall medical costs continue to rise rapidly, 2.92 times from 2012 to 2022 [3].

**Data availability statement:** All relevant data are within the manuscript and its Supporting Information files.

**Funding:** This study was financially supported by the National Social Science Fund of China (NSSFC) in the form of a grant (22BJY248). No additional external funding was received for this study. The funder had no role in study design, data collection and analysis, decision to publish, or preparation of the manuscript.

**Competing interests:** NO authors have competing interests.

To alleviate the health economic burden and to achieve equity in healthcare, China's healthcare system has undergone long-term progressive reforms, culminating in a system primarily centered around social health insurance, complemented by medical assistance, public healthcare, and commercial health insurance. Social health insurance comprises Urban Employee Basic Medical Insurance (UEBMI), Urban Resident Basic Medical Insurance (URBMI), and New Cooperative Medical Scheme (NCMS). Among these, UEBMI, initiated in 1999, addresses the healthcare needs of urban employed and retired personnel. Subsequently, the Chinese government introduced NCMS and URBMI in 2003 and 2007, respectively, achieving a 95% coverage rate for basic health insurance. In particular, these initiatives have improved the accessibility and affordability of healthcare services for the entire population, thereby alleviating some of the challenges associated with high medical costs and difficulties in accessing healthcare for Chinese residents [4].

The establishment of the three basic health insurance systems has gradually narrowed the gap in healthcare spending between urban and rural areas. However, due to the existence of constraints such as the urban-rural income gap, the difference in medical resources between urban and rural areas, and the urban-rural institutional division, inequality in access to medical care still exists between urban and rural areas in China [5]. Due to income constraints, rural residents cannot afford to see a doctor, and the rate of residents not seeking medical treatment for two-week illnesses is as high as 37.8% [6], which depresses their effective demand for medical consumption. There is still a gap between urban and rural residents in terms of the level of medical insurance benefits [7]. NCMS and URBMI exhibit notable variations in contribution standards, reimbursed medications, and designated hospitals (S1 Table). Integrating medical insurance becomes imperative to ensure fair medical coverage for both urban and rural residents.

To provide equitable access to medical care for urban and rural residents, the Chinese government needs to establish a unified and centralized social medical insurance system [8]. At present, Chinese scholars have three representative paths for the integration of basic medical insurance, the first of which is to merge the New Rural Cooperative and the Urban Residents' Medical Insurance to form basic medical insurance for residents, and then merge it with the basic medical insurance for urban workers, and finally to establish a nationwide national health insurance [9]. The second is to make it mandatory for all people to enroll in urban and rural health insurance, and then integrate urban and rural medical insurance when the urbanization rate reaches 70% to 80% [10]. The third is to abolish the urban workers' health insurance, so that all citizens will participate in the national health insurance as unified residents, with the same level of individual contributions and government subsidies [11]. However, the second path is limited by the differences between urban and rural areas, and it is difficult to realize the integration of the system in the short term. The third path can quickly achieve equal participation in the system, but the equalization of contributions fails to solve the problem of inequality caused by individual differences [12]. As the medical insurance systems for urban and rural residents are more homogeneous and closer in terms of funding levels, financial subsidies, and

reimbursement rates, they are easier to integrate and can solve the problems of double participation and double subsidies. Therefore, Chinese scholars generally agree that the integration of urban and rural residents' medical insurance should be realized first [13]. In January 2016, the State Council issued the "Opinions on Integrating the Basic Medical Insurance Systems for Urban and Rural Residents" [14], which started with the integration of the urban and rural medical insurance systems, integrating China's basic medical insurance and taking the first step in the 'three-step strategy'.

The URBMMI extends coverage to all urban and rural residents except those insured under the employees' basic medical insurance scheme and has expanded the scope of medical care (S2 table), stimulating demand for medical services [4]. Studies have shown that the URRBMI policy reduces the proportion of rural residents who are "in need but without access" [15], and there has also been a significant increase in the utilization of hospitalization services [16]. Unified facility management increases healthcare institutions, expanding accessible designated hospitals, improving medical care convenience, and reducing opportunity costs. After integration, original new rural cooperative medical care participants experienced a 108.51% funding increase, a 163.63% growth in reimbursed drugs, and a 50.84% rise in per capita fund expenditure [17]. Moreover, the level and scope of designated medical institutions are higher, which has significantly improved the accessibility and health of the insured (S2 Table).

Although health insurance helps to diversify the risk of illness as well as release medical needs, it can also raise ex-post facto moral hazard [18]. Ex-post moral hazard includes the moral hazard of participants and supply-side induced demand. In the process of integrating health insurance for urban and rural residents, the increase in the level of treatment induces participants to use more health care resources. Meanwhile, healthcare organizations are "two-way agents" that not only have an informational advantage in the process of patient care but are also the target of insurance payments. Medical institutions have the incentive to increase unnecessary medical services to obtain more insurance payments or health insurance funds [19,20]. Under these circumstances, while the integration policy releases the demand for healthcare, it also amplifies the moral hazard and induces higher growth rates in healthcare costs.

Existing literature has explored the impact of social medical insurance on healthcare utilization [21], medical costs [22], and health outcomes [23,24]. Studies have shown that health insurance incentivizes health care attendance and increases health care costs. Studies on the URRBMI integration indicate that integration reduces regional disparities in healthcare utilization [25] and alleviates unfairness in outpatient benefits [26]. However, debates persist regarding the impact of integration on medical costs. Some argue that integration, by enhancing protection levels, reduces medical costs, easing the inpatient costs for rural patients [27]and severe mental illness patients [28]. Conversely, another perspective suggests that integration increases medical costs. This is attributed to the release of medical demand during the URRBMI integration process, leading to heightened healthcare service utilization [17,29]. The controversy is whether policy studies distinguish between taking into account moral hazard effects. Given that China's URRBMI integration is in its early stages, and healthcare resource allocation efficiency is relatively low, it is essential to include a moral risk assessment when analyzing the cost-benefit of medical insurance reform [30]. Based on the above analysis, this paper formulates the hypothesis: the hypothesis is that urban-rural health insurance coordination increases the utilization behavior of healthcare resources by releasing residents' demand for healthcare and triggering moral hazard, thus increasing residents' medical costs. In addition, some of the studies have only examined the effect of health insurance on whether or not to seek medical care and the number of visits and to a lesser extent, the effect of health insurance on a patient's choice of provider level [31]. In China, more and more patients are concentrated in large hospitals, posing a serious challenge to the medical burden [32]. Therefore, a thorough investigation into the specific impact pathways of medical insurance integration on cost increases is necessary. Existing studies have mainly used the timing of provincial integration as a proxy for the timing of the opening of URBMMI in individual cities within each province. However, as the implementation of the policy is gradual, using the timing of provincial integration may lead to biased policy evaluation. To solve this problem, this paper matches the actual opening time of URRBMI in each city, and the results of the policy evaluation are more accurate.

The purpose of this paper is to assess the policy performance of China's implementation of medical insurance for urban and rural residents, including exploring the impact of the urban-rural medical insurance integration on the utilization of healthcare resources and healthcare costs of Chinese residents and explaining it from the perspectives of both demand release and moral hazard. This study utilizes CHARLS data from 2013, 2015, 2018, and 2020, employing Fixed-effects difference-in-differences model (Fixed-effects DID) and Propensity Score Matching difference-in-differences model (PSM-DID) methods to investigate the impact of China's URRBMI integration policy on medical costs for urban and rural residents. Objectives include:(1) This study uses a fixed-effects DID model to examine the overall impact of URRBMI integration on healthcare resource utilization and medical costs. It also explains the increase in medical costs in terms of both unlocking medical demand and moral hazard. (2) This study uses the PSM-DID model to address the endogeneity problem and demonstrate more rigorously that URRBMI integration raises the ex-post moral hazard problem. (3) This study examines the impact of URRBMI integration on residents' health awareness and preventive behavior and verifies whether integration causes the ex-ante moral hazard problem. The contributions are to (1) refine the framework for analyzing the URRBMI integration impact on healthcare resource utilization and medical costs, explain the reasons for the increase in medical costs in terms of medical demand release and moral hazard, and discuss whether the URRBMI integration can mitigate medical costs in the long term by promoting preventive behaviors. (2) Improve the accuracy of policy evaluation by using integrated data at the city level. Findings will guide China in achieving universal medical coverage, and lessons learned will benefit other developing countries.

## Methods

### Data source and study sample

Data were obtained from the China Health and Retirement Longitudinal Study (CHARLS), which has conducted five rounds of investigations. This study utilized data in 2013, 2015, 2018, and 2020. Data for 2011 was not used because only a few provinces implemented reforms to integrate URRBMI in 2011. The CHARLS sample is broadly representative, and the data contain detailed information on individual characteristics, healthcare behaviors, and healthcare costs, which fits the theme of this study. In addition, the sample has a low rate of lost visits [26,33,34].

This study employs a quasi-natural experiment to investigate the impact of URRBMI integration on healthcare utilization and costs. S3 Table outlines the implementation times of URRBMI integration in various cities across China. Unlike previous studies that often used provincial-level policy variables due to limited early-stage policy data, this study utilizes the specific policy implementation times of cities where CHARLS samples are located. Information regarding regional medical insurance integration policies was collected from local Human Resources and Social Security Bureau websites. It's important to note that the policy times used in this study represent the actual implementation times in each city rather than the policy introduction times.

We collated and integrated the data for 2013, 2015, 2018, and 2020, and then excluded 6734 samples that were insured with basic urban workers' health insurance, 7167 samples that purchased other medical insurance and private medical insurance, and excluded samples that were missing key individual characteristic scalars such as age and gender to obtain a sample of 58,670 observations (S1 Fig), with 25,670 in the experimental group and 33,000 in the control group (S1 Fig).

### Variables and measurement

**Measurement of medical resource utilization and medical cost.** The core dependent variables in this study are healthcare resource utilization and medical costs. Healthcare resource utilization comprises outpatient visits and inpatient visits. Outpatient visits measure "whether the individual had outpatient visits in the last month," and Inpatient visits measure "whether the individual had hospitalization in the last year." Additionally, it includes the hospital type for

both outpatient and inpatient care. This study categorizes hospital type into three types: Primary healthcare institutions (including health service stations, village clinics/private clinics, nursing homes, and community health service centers), secondary healthcare institutions (township health centers), and tertiary healthcare institutions (comprehensive hospitals, specialized hospitals, and traditional Chinese medicine hospitals). Data for 2013 and 2015 do not include nursing facilities; only 2018 and 2020 data include nursing facilities.

Medical costs were measured using out-of-pocket (OOP) costs for outpatient and inpatient care [30] and per capita family medical expenditure. outpatient OOP costs represent the participant's actual payments for outpatient services in the last month after deducting reimbursed amounts. Inpatient OOP costs measure the participant's actual payments for hospitalization in the past year after deducting reimbursed amounts. Medical expenditure includes both direct and indirect medical expenditures of the family. Indirect medical expenditures refer to the costs of traveling, nutrition, and accompanying family members due to medical treatment. The medical consumption variable, utilized in this study, is derived by dividing total family medical consumption by the family's population. Logarithmic transformations were applied to all medical cost variables in this study.

**Measurement of control variables.** Grossman's (2017) health needs model proposes that reasons such as age, income, education, uncertainty of disease, and price of healthcare services affect individual health needs [35]. Referring to the study of Zhu & Wang (2021), when exploring the impact of URRBMI integration on residents' medical costs, residents' demographic characteristics, health variables, health risk awareness, and household income variables should be included [28]. Combined with the information from the CHARLS database, individual characteristics in this study include age, gender, and marriage, health features involving health status and disability status, health awareness comprises smoking, drinking, regular check-ups, and physical exercise, income characteristics represent the participant's total income in the past year, combining annual wage, self-employment income, pension income, agricultural income, and personal business income. Logarithmic transformations were applied to income variables in this study (S4 Table).

## Model specifications

This study employed the DID Model to analyze China's URRBMI policy impact on residents' healthcare resource utilization and medical costs. The double difference method was first applied to the field of economics by Ashenfelter (1978) [36], and Chinese scholars Zhou and Chen (2005) [37] introduced the DID model to China and applied it to the evaluation of public policies. The DID model estimates the average treatment effect of a policy or intervention by comparing the difference between the treatment and control groups at two points in time before and after the policy or intervention. Assessing the policy effect under the traditional method is mainly done by setting a dummy variable for whether the policy occurs or not and then regressing it, in contrast, the model setup of the DID method is more scientific and can estimate the policy effect more accurately, and it can also avoid the problem of endogeneity to a certain extent. The nationwide implementation of the URRBMI integrated policy in 2016 served as a quasi-natural experiment. The study considered 2016 as the policy implementation year, using data from 2013 and 2015 as pre-policy samples and data from 2018 and 2020 as post-policy samples. Residents in URRBMI or NCMS in 2013 and 2015 who continued with URRBMI in 2018 and 2020 formed the experimental group, while those in URBMI or NCMS in 2013 or 2015, not join URRBMI during the study period, comprised the control group.

This paper further controls for time and area factors in the DID model. Referring to the research of He & Shen (2021), Shen (2022), and Pan et al. (2013) [38–40], region-fixed effects were included to control for region characteristics that did not change over time. As CHARLS is a long-term tracking data, this study also draws on Rubin (1974) and uses the time-fixed effects model [41], which has the advantage of excluding the effect of unobservable factors that are constant over time and controlling for changes in the year of URBMMI implementation. Gardiner & Luo (2010) point out that fixed-effects models control for individual trends over time and are more realistic, whereas random-effects models while controlling for variation between data, do not identify all differences between individuals and therefore underestimate standard errors

[42]. Meanwhile, most of the models related to health insurance policy evaluation by scholars such as Ren et al. (2022), Li et al. (2023), and Huang &Wu (2020) used fixed effects models [26,30,32]. Therefore, this study adopted a fixed-effects double-difference model in the assessment concerning the integration of urban and rural residents' medical insurance and residents' medical costs. In the identification of the integration time, this study uses the data of URRBMI integration at the municipal level, compared to previous studies that use the provincial integration time, this paper helps to control for the fact that the integration policy implementation will be different at the local level, which is important for the evaluation of the local policy.

Drawing on model set by Li et al. (2023) and Huang & Wu (2020) [30,32], the model is set as follows:

$$U_{it} = \beta_0 + \beta_1 DID_{it} + \beta_2 Treatment_i + \beta_3 Post_t + \sum_m \alpha_m X_{it}^m + \lambda_{it} + \mu_{it} \tag{1}$$

$$DID_{it} = Treatment_i \times Post_t \tag{2}$$

In the equation (1), subscripts $i$ and $t$ represent individual and time, respectively. The dependent variable $U_{it}$ it is the healthcare service utilization or medical costs of individual $i$ in period $t$, including outpatient OOP costs, inpatient OOP costs, outpatient visits, and inpatient visits as alternative variables. If individual $i$ participates in URRBMI in period $t$, it is set as 1. $DID_{it}$ is the core explanatory variable, in the form shown in equation (2), and is jointly determined by $Treatment_i$ and $Post_t$. $\beta_1$ is the coefficient corresponding to the core explanatory variable $DID_{it}$, also the focus of this study. If $\beta_1 >$ 0, it indicates that URRBMI integration increases individual medical service utilization or raises medical costs, and vice versa. $Treatment_i$ is a grouping dummy variable; if $i$ belongs to the experimental group, $Treatment_i$ is set as 1; otherwise, if $i$ belongs to the control group, $Treatment_i$ is set as 0. $Post_t$ is a period dummy variable; if $i$ is in the post-URRBMI integration experimental period (2018), $Post_t$ is set as 1; otherwise, $Post_t$ is set as 0. The coefficient $\beta_2$ of the grouping dummy variable $Treatment_i$ represents the difference between the experimental and control groups, indicating the existence of this difference even if the URRBMI policy is not implemented. The coefficient $\beta_3$ of the period dummy variable $Post_t$ represents the inherent difference between the two periods, even if the URRBMI policy is not implemented, indicating common time trends.

This study defines $X_{it}^m$ as control variables, mainly encompassing individual characteristics such as residents' age, gender, and marriage; health status variables including health status and disability status; health awareness variables like drinking, smoking, regular medical checkups, and physical exercise; and individual annual income. $\mu_{it}$ represents time-fixed effects. $\lambda_{it}$ represents region-fixed effects.

The core idea of the PSM method is to estimate the policy effect very effectively by calculating the propensity scores of the treatment group and the control group and then performing kernel matching or other matching methods to find the individuals in the control group who are most similar to the treatment group so that the two groups of data have similar characteristics in terms of variable observations, which can exclude the influence of other factors (Keisuke et al., 2003) [43]. This study firstly uses the PSM method to match the individual characteristics of residents, controlling for the effect on the demand for healthcare because of their characteristics such as age, gender, income, and education, and then evaluates using the DID method, which aims to remove the effect of the release of healthcare demand on the increase in healthcare costs and to identify the phenomenon of induced demand in the integration.

## Results

### Descriptive statistics

Table 1 calculates the changes and significance of between-group differences in the sample core variables before and after participation in URRBMI in 2013–2015 and 2018–2020, mean (scale) and t-test were used for continuous variables;

**Table 1. Descriptive statistics of the dependent and independent variables (N = 58670).**

| Variables | 2013-2015 (N = 25670) | | 2018(N = 33000) | | Change | P value |
|---|---|---|---|---|---|---|
| | Control | Treatment | Control | Treatment | | |
| | (N = 5622) | (N = 20048) | (N = 7220) | (N = 25780) | | |
| Age(mean,SD) | 61.17(9.89) | 61.03(9.85) | 62.81(9.99) | 62.74(9.87) | 1.7 | <0.001 |
| Gender | | | | | | 0.03 |
| Male | 2,948 (52.4) | 10,741 (53.6) | 3797 (53.8) | 13801(54.8) | 1.1 | |
| Female | 2,674 (47.6) | 9,307 (46.4) | 3,263 (46.2) | 11,369 (45.2) | −1.1 | |
| Education | | | | | | <0.001 |
| High school and above | 137(6.51) | 220(6.71) | 452(8.04) | 763(7.84) | 1.1 | |
| Marriage | | | | | | <0.001 |
| Unmarried | 1187(12.67) | 2105(13.08) | 837(14.89) | 1452(14.64) | 1.8 | |
| Married | 8181(87.33) | 13988(86.92) | 4786(85.11) | 8307(85.36) | −1.8 | |
| Health status | | | | | | <0.001 |
| Very unhealthy | 245(5.36) | 536(6.76) | 269(5.15) | 622(6.82) | 0.0 | |
| Less healthy | 918(20.07) | 1894(23.6) | 976(18.68) | 2119(23.23) | −0.9 | |
| General healthy | 2293(50.12) | 3870(48.83) | 2501(47.87) | 4455(48.85) | −0.8 | |
| More healthy | 568(12.42) | 933(11.77) | 680(13.01) | 1007(11.04) | −0.2 | |
| Very healthy | 551(12.04) | 692(8.73) | 799(15.29) | 917(10.05) | 2.1 | |
| Disability | | | | | | <0.001 |
| No | 8352(89.15) | 14127(87.78) | 5177(92.07) | 8804(90.46) | 2.8 | |
| Yes | 1016(10.85) | 1966(12.22) | 466(7.93) | 928(9.54) | −2.8 | |
| Physical activity | | | | | | |
| No | 905 (37.9) | 2,737 (33.4) | 3,166 (44.9) | 9,857 (39.2) | −5.4 | |
| Yes | 1,481 (62.1) | 5,466 (66.6) | 3,880 (55.1) | 15,273 (60.8) | 5.4 | |
| Smoking | | | | | | <0.001 |
| No | 3024(59.83) | 5150(60.34) | 5724(98.33) | 9398(98.41) | 38.2 | |
| Yes | 2030(40.17) | 3385(39.66) | 92(1.67) | 152(1.59) | −38.2 | |
| Drinking | | | | | | 0.05 |
| No | 6958(74.50) | 11717(72.98) | 4241(75.57) | 7170(73.76) | 0.9 | |
| Yes | 2382(25.50) | 4337(27.02) | 1371(24.43) | 2551(26.24) | −0.9 | |
| Regular medical checkups | | | | | | |
| No | 7186(76.71) | 12770(79.35) | 3920(69.71) | 7034(72.28) | −7.1 | |
| Yes | 2182(23.29) | 3323(20.65) | 1703(30.29) | 2698(27.72) | 7.1 | |
| Outpatient visits | | | | | | <0.001 |
| No | 4,554 (81.1) | 15,680 (78.4) | 5,826 (82.6) | 20,451 (81.3) | −2.0 | |
| Yes | 1,058 (18.9) | 4,317 (21.6) | 1,228 (17.4) | 4,695 (18.7) | 2.0 | |
| Type of outpatient | | | | | | <0.001 |
| Primary | 753(39.76) | 1565(41.57) | 219(27.14) | 505(29.92) | −12.0 | |
| Secondary | 498(26.29) | 986(36.19) | 222(27.51) | 502(29.74) | 2.8 | |
| Tertiary | 643(33.95) | 1214(32.24) | 366(45.35) | 681(40.34) | 9.2 | |
| Inpatient visits | | | | | | |
| No | 8198(87.63) | 13865(86.23) | 4783(85.09) | 8019(82.42) | −3.3 | |
| Yes | 1157(12.37) | 2214(13.77) | 838(14.91) | 1711(17.58) | 3.3 | |
| Type of inpatient | | | | | | |
| Primary | 18(1.87) | 47(2.54) | 2(0.28) | 6(0.41) | −1.9 | |
| Secondary | 185(19.23) | 384(20.71) | 103(14.45) | 278(19.16) | −2.6 | |

*(Continued)*

**Table 1.** (Continued)

| Variables | 2013-2015 (N = 25670) | | 2018(N = 33000) | | Change | P value |
|---|---|---|---|---|---|---|
| | Control | Treatment | Control | Treatment | | |
| | (N = 5622) | (N = 20048) | (N = 7220) | (N = 25780) | | |
| Tertiary | 759(78.90) | 1423(76.75) | 608(85.27) | 1167(80.43) | 4.5 | |
| Outpatient OOP costs (mean,SD) | 1219.88(4657.82) | 1116.00(3649.58) | 1519.28(7033.70) | 1476.57(5537.69) | 340.5 | 0.014 |
| Inpatient OOP costs(mean,SD) | 17935.94(25678.61) | 15278.68(21452.04) | 22420.22(36771.27) | 21305.95(30649.66) | 5661.3 | <0.001 |
| Outpatient costs (mean,SD) | 1219.88(4657.824) | 1115.99(3649.58) | 1519.28(7033.70) | 1476.57(5537.69) | 340.0 | 0.044 |
| Inpatient costs(mean,SD) | 17635.94(25678.61) | 15278.68(21452.04) | 22420.22(36771.27) | 21305.95(30649.66) | 5661.0 | <0.001 |
| Medical expenditure(mean,SD) | 3646.266(12775.22) | 4038.85(11470.69) | 9411.15(39266.25) | 6629.92(20429.33) | 3289.1 | <0.001 |
| Other expenditure | 5339.81(28934.55) | 5224.65(20015.98) | 10707.80(41648.13) | 9486.33(55894.16) | 4475.0 | <0.001 |
| Distance to medical institutions | 27.34(49.00) | 44.30(266.49) | 34.67(105.26) | 58.98(246.29) | 12.4 | 0.047 |
| Satisfaction with medical services | | | | | | |
| Very dissatisfied | 290(6.42) | 562(7.45) | 403(7.42) | 902(9.61) | 1.7 | |
| More dissatisfied | 497(11.00) | 936(12.40) | 361(6.65) | 695(7.41) | −4.8 | |
| Generally satisfied | 1762(39.00) | 3134(41.52) | 2403(44.24) | 4265(45.46) | 4.4 | |
| More satisfied | 1147(25.39) | 1755(23.25) | 1338(24.63) | 2079(22.16) | −1.0 | |
| Very satisfied | 822(18.19) | 1161(15.38) | 927(17.07) | 1441(15.36) | −0.4 | |
| Income | 16688.24(26208.28) | 16357.81(40346.37) | 23974.92(108657.4) | 20339.09(47763.25) | 5175.4 | <0.001 |

*Note.* Changes and P values were calculated for all respondents before and after URRBMI integration in the years of 2013–2015 and 2018–2020. Mean (SD) and t test were conducted for continuous variables; n (%) and chi-square test were conducted for categorical variables.

n (%) and chi-square test were used for categorical variables. The results in Table 1 show that the age of the sample in the experimental period increases significantly compared to the base period, which may be because senior patients are more inclined to purchase health insurance and the URRBMI policy improves the level of treatment and attracts more residents, especially senior residents, to purchase health insurance. The results in Table 1 also show that the proportion of the sample in the experimental period that is educated and married has significantly increased, which may be related to the fact that residents who are educated and married have higher health awareness and therefore purchase more urban and rural health insurance. Meanwhile, the health level of the sample in the experimental period increased significantly, which may be attributed to the health effect of the URRBMI.

 Among the core variables, the probability of physical exercise, the proportion of non-smokers, and the proportion of those who participated in routine medical checkups increased in the sample in the experimental period, suggesting that the URRBMI may have increased the health awareness of the sample. Meanwhile, the probability of the sample's outpatient visit increased in the experimental period, and the Inpatient OOP cost and medical consumption increased significantly first, suggesting that the URRBMI may have increased residents' medical costs by increasing their healthcare resource utilization behaviors. Finally, the proportion of the sample going to primary and secondary hospitals decreased while the proportion going to senior hospitals increased in the experimental period, suggesting that the URRBMI may have led to a moral hazard problem by encouraging the residents to go to a higher level of healthcare institutions, which in turn may have led to an increase in medical cost. Further modeling is needed to precisely ascertain the specific impact of URRBMI integration on urban and rural residents' medical costs.

## Impact of URRBMI integration on medical resource utilization and medical costs

The results of the descriptive analyses in Table 1 indicate that residents' healthcare resource utilization and medical costs have risen after the opening of URRBMI integration in most of China's cities. To verify the policy effect of medical insurance integration more rigorously, this part uses the DID method to conduct empirical tests. Table 2 presents the impact of URRBMI integration on outpatient visits, inpatient visits, outpatient OOP costs, inpatient OOP costs, and medical consumption. Models (1)-(5) report the results of DID analysis (S5 Table), while models (6)-(7) present the results of the DID with fixed-time-effects and fixed-region-effects (S6 Table). The findings indicate that the implementation of the URRBMI policy significantly increases the probability of hospitalization and raises outpatient OOP costs, inpatient OOP costs, and medical consumption. After the implementation of the URRBMI integration, the probability of residents' inpatient visits, outpatient OOP costs, inpatient OOP costs, and medical consumption rose by 1.9, 22.1, 24.7, and 167.8 percentage points, respectively. These results remain significant even after introducing time and region control variables. Therefore, the implementation of the URRBMI policy has, to a certain extent, unleashed the suppressed healthcare demand, especially in terms of hospitalization, leading to an increase in medical costs.

Personal characteristics control for sex, age, marriage. Health characteristics control for self-rated health, disability. Health awareness controls smoking, drinking, medical checkups. Income characteristics control for annual personal income.

## Sources of increased medical costs: demand release or moral hazard

The above analyses show that the URRBMI integration has significantly increased the probability of consultation and medical costs, however, they cannot explain whether the increase in medical costs comes from the release of residents' normal medical demands or moral hazards.

Since URRBMI integration does not directly give participants subsidies to increase their income, the income and consumption levels of participants are hardly affected by the URRBMI integration policy. Therefore, to explore whether the

Table 2. Impact of URRBMI integration on healthcare resource utilization and medical costs.

| Variables | DID | | | | | Fixed Effects Model | | | | |
|---|---|---|---|---|---|---|---|---|---|---|
| | Outpatient visits | Inpatient visits | Outpatient OOP costs | Inpatient OOP costs | Medical expenditure | Outpatient visits | Inpatient visits | Outpatient OOP costs | Inpatient OOP costs | Medical expenditure |
| DID | 0.000 | 0.019*** | 0.152** | −0.001 | 0.261 | 0 | 0.019* | 0.221** | 0.247** | 1.678*** |
| | (0.001) | (0.005) | (0.074) | (0.086) | (0.551) | (0.002) | (0.011) | (0.085) | (0.120) | (0.629) |
| Personal characteristic | YES | YES | YES | YES | YES | YES | YES | YES | YES | YES |
| Health characteristics | YES | YES | YES | YES | YES | YES | YES | YES | YES | YES |
| Health awareness | YES | YES | YES | YES | YES | YES | YES | YES | YES | YES |
| Income characteristics | YES | YES | YES | YES | YES | YES | YES | YES | YES | YES |
| Time effect | NO | NO | NO | NO | NO | YES | YES | YES | YES | YES |
| Region effect | NO | NO | NO | NO | NO | YES | YES | YES | YES | YES |
| Observations | 21047 | 21033 | 1710 | 1266 | 4676 | 21060 | 21046 | 1714 | 1272 | 4683 |
| R-sq | 0.037 | 0.077 | 0.033 | 0.087 | 0.033 | 0.038 | 0.078 | 0.141 | 0.228 | 0.044 |

*Note.* *, **, *** corresponding to p values ≤ 0.10, ≤ 0.05 and ≤ 0.01, respectively. 95% confidence interval reported in brackets.

increase in medical costs comes from the release of individual healthcare demand or moral hazard, it can be tested by distinguishing the crowding-out effect of medical expenditure from the expenditure of other expenditures by participants with different abilities to pay. If medical expenditure does not have a crowding-out effect on other expenditures, it means that medical expenditure is within their affordability and the increase in medical expenditure brought about by URRBMI integration comes more from moral hazard. If medical expenditure has a crowding-out effect on other expenditures, then the URRBMI integration mainly releases the suppressed medical demand of the insured.

Table 3 reports the results of the fixed-effects DID model, which shows that URRBMI integration significantly raises inpatient costs for the low- and middle-income groups and that medical expenditure crowds out other consumption. For the high-income group, URRBMI integration significantly raises outpatient and inpatient costs but has no crowding-out effect on other expenditures. This suggests that URRBMI integration releases the medical demand of participants, mainly for low- and middle-income participants. For participants with high income levels, the impact of URRBMI integration on their medical costs is mainly in the form of a moral hazard.

Statistics from China's health sector show that in 2017, the total number of outpatient visits nationwide increased by 250 million over the previous year, an increase of 3.2%, the average number of visits by residents increased from 5.8 in 2016 to 5.9 in 2017, the total number of hospitalizations nationwide increased by 17.08 million over the previous year (an increase of 7.5%), and the annual hospitalization rate increased from 16.5% in 2016 to 17.6%. Meanwhile, Xiang et al.(2020) [44] argued that due to the strong monopoly of China's public medical institutions and the lack of effective tools to constrain and monitor medical behavior, the increase in the rate of medical insurance integration and compensation, the moral hazard of the residents has instead intensified, and at the same time, doctors can induce overmedication of their patients by providing more medical services, changing the type of medical services or breaking down hospitalization. The above evidence suggests that demand release effects and moral hazard problems do exist in the integration of URRBMI in China, consistent with the results of this study.

**Table 3. Impact of URRBMI integration on medical and other expenditures of different income groups.**

| Variables | Lower middle income group | | | High income group | | |
|---|---|---|---|---|---|---|
| | Outpatient costs | Inpatient costs | Percentage of other consumption | Outpatient costs | Inpatient costs | Percentage of other consumption |
| DID | 0.093 | 0.257* | −0.049*** | 0.409*** | 0.096* | 0.035 |
| | (0.129) | (0.155) | (0.017) | (0.141) | (0.120) | (0.018) |
| Personal characteristic | YES | YES | YES | YES | YES | YES |
| Health characteristics | YES | YES | YES | YES | YES | YES |
| Health awareness | YES | YES | YES | YES | YES | YES |
| Income characteristics | YES | YES | YES | YES | YES | YES |
| Time effect | YES | YES | YES | YES | YES | YES |
| Region effect | YES | YES | YES | YES | YES | YES |
| Observations | 876 | 671 | 2959 | 853 | 657 | 3070 |
| R-sq | 0.122 | 0.147 | 0.125 | 0.199 | 0.299 | 0.148 |

*Note.* *, **, *** corresponding to p values ≤ 0.10, ≤ 0.05 and ≤ 0.01, respectively. 95% confidence interval reported in brackets.

## Endogeneity test for moral hazard

The previous section verified that there is both demand release and moral hazard in the process of medical insurance integration, and to more rigorously verify the impact of medical insurance integration on the moral hazard of residents and to mitigate the endogeneity problem, this section utilizes the PSM-DID to conduct an empirical test. Referring to Pan (2022) [45], Stuart et al. (2014) [46], and Srensen & Grytten (1999) [47], to address the issue of endogeneity and to validate the existence of a moral hazard, this study adopted the propensity score matching (PSM) method in the subsequent analyses. K-nearest neighbor matching (1:3 ratio) identifies control group residents with similar individual characteristics to those in the experimental group, controlling for residents' health status, health awareness, and income in assessing the impact of URRBMI integration on medical costs. S2–S4 Figs illustrate the dispersion of control variables before and after matching, with outpatient OOP costs, inpatient OOP costs, and medical consumption as dependent variables. The reliability of PSM depends on the fulfillment of the 'conditional independence condition', which requires that there is no significant difference between the samples of the treatment and control groups in terms of the observable variables after matching, and the results of Rosenbaum & Donald (1983) [48] show that matching is considered to be ineffective when the absolute value of the standard deviation of the matched variables is greater than 20. From the results of Tables S7–S12, it can be seen that compared to the pre-matching period, the differences between the matched treatment and control groups in terms of age, gender, marriage, routine physical examination, health status, disability, alcohol consumption, smoking, and income level have decreased significantly, and the absolute value of the standard deviation of each matching variable is significantly less than 10. From the value of the probability of matching in the t-test of the means, it can be seen that after matching, there is no significant difference in all the variables of the matched treatment and control groups. Control group does not have significant differences in all variables, and the overall matching results are good. Therefore, the kernel matching estimation used in this study is reliable and can be used for the next step of double difference analysis.

Table 4 presents the results of the PSM-DID method (S13 Table). It is evident that, even with minimal differences in residents' demographic characteristics, health status, health risk awareness, income variables, and healthcare resource disparities, URRBMI integration still significantly increases residents' outpatient OOP costs, inpatient OOP costs, and medical consumption. This indicates that URRBMI integration not only releases the residents' demand for healthcare but also promotes moral hazard in the process of accessing healthcare.

Table 4. Impact of URRBMI integration on moral hazard.

| Variables | (1) | (2) | (3) |
|---|---|---|---|
| | Outpatient OOP costs | Inpatient OOP costs | Medical expenditure |
| DID | 0.237*** | 0.282* | 0.384*** |
| | (0.085) | (0.152) | (0.076) |
| Personal characteristic | YES | YES | YES |
| Health characteristics | YES | YES | YES |
| Health awareness | YES | YES | YES |
| Income characteristics | YES | YES | YES |
| Region effect | YES | YES | YES |
| Time effect | YES | YES | YES |
| Observations | 1577 | 927 | 4593 |
| R-sq | 0.145 | 0.303 | 0.069 |

*Note.* *, **, *** corresponding to p values ≤ 0.10, ≤ 0.05 and ≤ 0.01, respectively. 95% confidence interval reported in brackets.

## Impact of URRBMI integration on utilisation utilization of different types of medical resources

The results of the descriptive analyses in Table 1 indicate that the proportion of residents choosing to visit higher-level medical institutions increases after the integration of urban and rural residents' medical insurance. To more rigorously verify the effect of medical insurance integration on residents' choice of consultation after developing moral hazard problems, this part uses the PSM-DID method to conduct an empirical test. Table 5 further illustrates the impact of moral hazard in the integration process on the residents' choice of hospital type for medical care. Using the post-propensity score-matched data, a fixed-effects model was employed to explore URRBMI integration's effect on hospitalization levels. The results indicate a significant increase in residents' outpatient and inpatient visits to higher-level hospitals post-integration, the type of outpatient visit and the type of hospitalization of the population rose by 16.3% and 4.9%, respectively. This suggests that URRBMI integration induces residents to seek treatment in higher-level hospitals, supported by a notable increase in hospital distance, affirming this phenomenon (S14 Table).

## Impact of URRBMI integration on preventive behaviors

Table 6 presents the impact of URRBMI integration on residents' health awareness and preventive behaviors (S15 Table). Post-integration, residents increased engagement in physical exercise and significantly reduced smoking habits, the proportion of the population participating in physical exercise increased by 19.3% and the proportion of smokers decreased by 35.7%. Additionally, integration resulted in a higher prevalence of regular health check-ups among residents, the probability of the population attending routine medical check-ups has risen by 6%, contributing positively to health improvement. However, in the short term, these actions have not yet translated into reduced healthcare demands and subsequent decreases in medical costs. Despite the initial short-term increase in healthcare costs following URRBMI integration, the long-term improvement in residents' health consciousness and preventive behaviors holds the potential to alleviate the burden on healthcare costs to some extent.

## Common trend and robustness analysis

The study's DID model passed the common trends test (S5 Fig, S16 Table). To ensure robustness, the sample over 65 and sample under 65 replaced those in Table 2, Tables 5–6 (S17–S22 Tables). In addition, to check the robustness of the PSM-DID results, we replaced the K-nearest neighbor matching ratio with 1:2. The results remain robust when grouped

**Table 5. Impact of URRBMI integration on type of medical institutions and distance to medical institutions.**

| Variables | (1) | (2) | (3) |
|---|---|---|---|
| | Outpatient type | Inpatient type | Distance to medical institutions |
| DID | 0.163*** | 0.049* | 29.864*** |
| | (0.049) | (0.025) | (9.072) |
| Personal characteristic | YES | YES | YES |
| Health characteristics | YES | YES | YES |
| Health awareness | YES | YES | YES |
| Income characteristics | YES | YES | YES |
| Region effect | YES | YES | YES |
| Time effect | YES | YES | YES |
| Observations | 3715 | 2791 | 2685 |
| R-sq | 0.100 | 0.057 | 0.022 |

*Note.* *, **, *** corresponding to p values ≤ 0.10, ≤ 0.05 and ≤ 0.01, respectively. 95% confidence interval reported in brackets.

**Table 6. Impact of URRBMI integration on health awareness and preventive behaviors.**

| Variables | (1) | (2) | (3) |
| --- | --- | --- | --- |
| | Physical exercise | Cigarette smoking | Regular medical checkups |
| DID | 0.193*** | −0.357*** | 0.060*** |
| | (0.019) | (0.012) | (0.014) |
| Personal characteristic | YES | YES | YES |
| Health characteristics | YES | YES | YES |
| Health awareness | YES | YES | YES |
| Income characteristics | YES | YES | YES |
| Region effect | YES | YES | YES |
| Time effect | YES | YES | YES |
| Observations | 17583 | 21057 | 21047 |
| R-sq | 0.07 | 0.322 | 0.077 |

*Note.* *, **, *** corresponding to p values ≤ 0.10, ≤ 0.05 and ≤ 0.01, respectively. 95% confidence interval reported in brackets.

according to age and when the matching method is changed (S23 Table). Finally, to exclude the effect of 'COVID-19', we further re-test the results in Table 2 (S24 Table) using the fixed-effects DID method with data from 2013−2018, and the results show that even after excluding the effect of 'COVID-19', the URRBMI integration still increases residents' consultation behavior and medical costs.

## Discussion

This study examines the impact of URRBMI integration on the healthcare costs of urban and rural residents and focuses on the impact of moral hazard in the integration process. This study innovatively examines the impact of URRBMI integration on the growth of healthcare costs from the perspectives of demand release and ex-post moral hazard and adds a discussion of ex-ante moral hazard. Unlike previous studies, this study uses city-level data to make the regression results more precise. It also distinguishes between hospital levels to make a more specific assessment of medical resource utilization. The results are summarized below: (1) The results of the DID analysis with double fixed effects show that although the integration of the URRBMI integration triggered an increase in medical costs, it also released the repressed medical demand of residents and improved the utilization of medical resources. (2) The sources of increase in medical costs are demand release and moral hazard. For the low- and middle-income groups, the URRBMI integration mainly triggers the increase of medical costs through demand release. For insured individuals in the high-income group, the URRBMI integration increases medical costs mainly through the ex-post moral hazard problem. (3) After controlling for residents' characteristics using the PSM-DID method, URRBMI integration still brings about an increase in medical resource utilization and medical costs, proving the existence of the moral hazard problem. Moreover, moral hazard encourages residents to use the medical resources of advanced medical institutions more than those of primary medical institutions, which leads to an increasing medical cost. (4) The URRBMI integration has increased residents' preventive behavior and health awareness. Integration did not trigger ex-ante moral hazard problems, which is similar to the findings of previous studies.

There are some shortcomings in this study. First, in terms of data span, limited by data availability, this study only assesses the impact of health insurance integration for urban and rural residents in China from 2013–2020 and does not assess the policy effects from 2020 to the present. This study verifies that health insurance integration may improve health in the long run by raising residents' health awareness and increasing preventive behaviors, which ultimately reduces healthcare costs, but it does not do so with relevant data. Second, in the identification of moral hazard, this study

only verified the existence of moral hazard in the process of health insurance integration but did not strictly distinguish whether the source of moral hazard is the moral hazard of individual residents or the induced demand of the supply side of the healthcare organization, which is a research direction that can be explored in the future. Finally, in terms of the general adaptation of the research object, this paper adopts CHRLS data to explore the healthcare resource utilization behavior of Chinese residents aged 45 and above. Although the middle-aged and elderly groups are the groups with the highest incidence of diseases, children, adolescents, and young people in China are also covered by healthcare insurance integration, and exploring the healthcare resource utilization and healthcare cost issues of these groups is also of some significance. Of course, this study is of positive significance to the improvement of China's urban and rural residents' medical insurance reform, and it is expected that the shortcomings of this study can be improved in future research.

## Conclusions

This study provides new evidence that URRBMI integration increases medical costs from a moral hazard perspective. Although the integration reform increased medical costs in the short run, it also had the following positive effects: (1) It released the suppressed medical needs of residents, which helped to improve the efficiency of healthcare resource utilization. (2) Raising the health awareness of residents, which may improve their health in the long run. Of course, there is a serious moral hazard problem in the integration process, which exacerbates the pressure on the utilization of medical resources in China.

The results of this study have important policy implications for the optimization of health insurance systems in developing countries such as China. On the one hand, it is necessary to take medical demand and moral hazard into account when assessing the effects of health insurance policies. On the other hand, integration has brought about the problem of residents crowding high-level hospitals, resulting in a waste of medical resources. In terms of health policy, promoting family doctor contracting services, setting up a reasonable cost-sharing mechanism, and reforming the health insurance payment method are feasible ways to mitigate moral hazard and reduce the medical burden.

In this regard, this study proposes the following future research directions: (1) Assess the long-term policy effects of integration reforms and explore whether China's integration reforms can improve the health of the population in the long run by raising health awareness, which ultimately reduces healthcare costs. (2) Exploring the sources of moral hazard arising from China's URRBMI integration. Explore the mitigation mechanism of moral hazard. Evaluate the role of China's hierarchical diagnosis and treatment system or payment reform in mitigating moral hazard; or how to adjust the level of medical insurance benefits to control the increase in healthcare burden.

## Supporting information

**S1 Table. Income and Expenditure of NCMS and URBMI in 2016.**
(DOCX)

**S2 Table. Policy changes after health insurance integration.**
(DOCX)

**S3 Table. Duration of urban and rural residents' health insurance integration by city.**
(DOCX)

**S4 Table. Meaning of variables.**
(DOCX)

**S5 Table. Impact of URRBMI on healthcare resource utilization and medical costs-DID.**
(DOCX)

**S6 Table. Impact of URRBMI on healthcare resource utilization and medical costs-DID with fixed time effects and fixed region effects.**
(DOCX)

**S7 Table. PSM matching results test 1 (explanatory variable is outpatient OOP costs).**
(DOCX)

**S8 Table. PSM matching results test 2 (explanatory variable is outpatient OOP costs).**
(DOCX)

**S9 Table. PSM matching results test 1 (explanatory variable is inpatient OOP costs).**
(DOCX)

**S10 Table. PSM matching results test 2 (explanatory variable is inpatient OOP costs).**
(DOCX)

**S11 Table. PSM matching results 1 (explanatory variable is medical expenditure).**
(DOCX)

**S12 Table. PSM matching results 2 (explanatory variable is medical expenditure).**
(DOCX)

**S13 Table. Impact of URRBMI integration on moral hazard.**
(DOCX)

**S14 Table. Impact of URRBMI integration on type of medical institutions and distance to medical institutions.**
(DOCX)

**S15 Table. Impact of URRBMI integration on health awareness and preventive behaviors.**
(DOCX)

**S16 Table. Joint significance test for leads and lags.**
(DOCX)

**S17 Table. Impact of URRBMI integration on healthcare resource utilization and medical costs for residents over 65 years of age.**
(DOCX)

**S18 Table. Impact of URRBMI integration on healthcare resource utilization and medical costs for residents under 65 years of age.**
(DOCX)

**S19 Table. Impact of URRBMI integration on type of medical institutions and distance to medical institutions for residents over 65 years of age.**
(DOCX)

**S20 Table. Impact of URRBMI integration on type of medical institutions and distance to medical institutions for residents under 65 years of age.**
(DOCX)

**S21 Table. Impact of URRBMI integration on health awareness and preventive behaviors for residents over 65 years of age.**
(DOCX)

**S22 Table. Impact of URRBMI integration on health awareness and preventive behaviors for residents under 65 years of age.**
(DOCX)

**S23 Table. Impact of URRBMI integration on induced demand- K-nearest neighbor matching (1:2 ratio).**
(DOCX)

**S24 Table. Policy effects of URRBMI integration after removing the impact of the COVID-19.**
(DOCX)

**S1 Fig. Sample selection and inclusion/exclusion criteria.**
(DOCX)

**S2 Fig. Degree of control variable dispersion before and after PSM matching.**
(DOCX)

**S3 Fig. Degree of control variable dispersion before and after PSM matching.**
(DOCX)

**S4 Fig. Degree of control variable dispersion before and after PSM matching.**
(DOCX)

**S5 Fig. Equilibrium trend test of medical costs.**
(DOCX)

## Acknowledgments

The authors give special thanks to the China Health and Retirement Longitudinal Study Project team and the survey respondents. We are very grateful to the reviewers for their excellent comments on the logic, data, modeling, methodology, and conclusions of the article, which were very helpful for us in improving our research.

## Author contributions

**Conceptualization:** Chen Liu, Qun Su.

**Data curation:** Chen Liu, Huaizhen Xing.

**Formal analysis:** Chen Liu, Meng Wang.

**Funding acquisition:** Chen Liu.

**Methodology:** Chen Liu, Qun Su, Meng Wang.

**Supervision:** Qun Su, Huaizhen Xing.

**Validation:** Qun Su, Meng Wang, Huaizhen Xing.

**Writing – original draft:** Chen Liu.

**Writing – review & editing:** Chen Liu, Qun Su.

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
