## [Decision Letter · Decision Letter 0]

PONE-D-24-17636How did the Urban and Rural Resident Basic Medical Insurance integration Affect Medical Costs? ——Evidence from ChinaPLOS ONE?

Dear Dr. Su,

Thank you for submitting your manuscript to PLOS ONE. After careful consideration, we feel that it has merit but does not fully meet PLOS ONE’s publication criteria as it currently stands. Therefore, we invite you to submit a revised version of the manuscript that addresses the points raised during the review process.

We look forward to receiving your revised manuscript.

Kind regards,

Rinshu Dwivedi, Ph.D.

Academic Editor

PLOS ONE

2. Thank you for uploading your study's underlying data set. Unfortunately, the repository you have noted in your Data Availability statement does not qualify as an acceptable data repository according to PLOS's standards. At this time, please upload the minimal data set necessary to replicate your study's findings to a stable, public repository (such as figshare or Dryad) and provide us with the relevant URLs, DOIs, or accession numbers that may be used to access these data. For a list of recommended repositories and additional information on PLOS standards for data deposition, please see https://journals.plos.org/plosone/s/recommended-repositories .

Additional Editor Comments:

Dear Authors,

The required number of comments and suggestions are available and through the paper is of contemporary importance, the major changes and modifications are needed. Kindly go through the comments and suggestions of the esteemed reviewers and incorporate the same for the revision of the manuscript.

Best Regards

Reviewers' comments:

Reviewer's Responses to Questions

**Comments to the Author**

1. Is the manuscript technically sound, and do the data support the conclusions?

Reviewer #1: Partly

Reviewer #2: Yes

2. Has the statistical analysis been performed appropriately and rigorously?

Reviewer #1: No

Reviewer #2: Yes

3. Have the authors made all data underlying the findings in their manuscript fully available?

Reviewer #1: No

Reviewer #2: Yes

4. Is the manuscript presented in an intelligible fashion and written in standard English?

Reviewer #1: No

Reviewer #2: Yes

Reviewer #1: 1. Abstract part needs revision, the result and conclusion part can be extended to provide better information to the readers.

2. The author’s mentioned the following statement, “However, Chinese scholars are divided on the specific path of integration.” Any specific reason; please make it clear for readers. Reference missing.

3. Authors mentioned, “Nationwide implementation marks a practical choice to pursue the integration path by starting with the integration of medical insurance for urban and rural residents”. The readers would have difficulty to extract what the authors try to say here.

4. Authors mentioned the statement “existing studies ignore the hospital level” Any specific reason; please make it clear for readers. Reference missing.

5. Hypothesis of the present study missing.

6. Research question and Aim of the study: the objective is not well phrased. They are not clear on what they are aiming to address. Clarity missing.

7. The present study “organized and integrated data from 2013, 2015, and 2018”. It seems outdated data and predictions/implementation may not reach the objectives.

8. DID abbreviation must be mentioned in Introduction section and authors mentioned in methodology session.

9. The authors used the following statement “There has also been a significant increase in the level of hospitals visited”. Any specific reason; please make it clear for readers.

10. Table 1. Result indicated age is statistically significant. Any specific reason; please make it clear for readers. The interpretation of Tables is not clear.

11. The authors did not mention anywhere details of “statistical methods were used in data and other analysis…” information throughout the manuscript. May it is a very important section for the study in terms of reader understands.

12. The fixed-effects model will calculate only within the data variation, what about between data?

13. Over all manuscript; flow, lack of interpretation and data analysis at tables including citation. For example: Ref: 40: “Rubin D B. Estimating causal effects of treatments in randomized and nonrandomized studies. Journal of educational Psychology.1974, 66(5): 688. And also Ref: 42-44”

14. Though the idea is noble and the framed objectives are interesting, the overall manuscript is mismatching with the discussed or mentioned methods and findings.

Reviewer #2: First and foremost, I would like to commend the authors for addressing such a significant topic. The paper effectively delineates of China's healthcare insurance systems, as well as the associated challenges of escalating medical costs and disparities in access. This contextual background provides a robust foundation for understanding the need for policy reform. The paper is well-written and supported by sound methodology and findings. However, I have a few suggestions that could further enhance the quality and reliability of the paper.

Introduction section:

1. Abstract: The size of the abstract is lengthy therefore required to reduce it.

2. Problem Statement: The introduction would benefit from a more explicit problem statement that clearly defines the specific research gap addressed by the study. Articulating the research problem more precisely would enhance clarity and provide clearer direction.

3. Information Flow: Consider relocating detailed statistics and comparisons to the results or discussion sections. Streamlining the introduction to focus on key issues and research gaps will improve readability and coherence.

4. Transition and Focus: Improve the transition between the discussion of insurance systems and the rationale for integration. A clearer connection between the historical context and the study’s focus on URRBMI integration will enhance the logical flow.

Methodological section

1. Control Variables: Justify the selection of specific control variables more thoroughly. Clarifying how these variables align with theoretical frameworks or previous literature will strengthen the rationale for their inclusion.

2. Bias and Data Handling: While excluding provincial-level data is mentioned, a more concise explanation of how city-level data mitigates potential biases could enhance understanding of the study's robustness.

3. Model Specification: Ensure that the equations provided in the model specification are accompanied by clear explanations of each term to make the methodology more accessible, especially for readers less familiar with econometric techniques.

4. Endogeneity Test: Further detail the matching process and any potential biases in matching to add depth

5. Discuss Potential Limitations: Address potential limitations more explicitly, such as the impact of unobserved variables or the generalizability of the results to other regions or populations.

Results section

1. Clarify Statistical Significance: - Ensure that statistical significance levels and effect sizes are clearly reported in all tables.

2. Enhance transitions between sections for better readability. For example, connecting descriptive statistics directly to impact analysis and then to the moral hazard discussion could create a more cohesive narrative.

3. Incorporate Visual Aids: Include charts or graphs in addition to tables to visualize trends and changes more effectively. A line graph showing changes in inpatient visits over time could illustrate trends more dynamically.

4. Sources of Increased Medical Costs: Provide more specific examples or scenarios of demand release versus moral hazard to make the discussion more tangible.

5. Impact of URRBMI Integration: Specify the magnitude of increases in hospitalization and medical costs and how they compare to baseline values.

Overall, the paper is well-structured, with a clear focus on evaluating the impact of URRBMI integration. Addressing the suggestions above can enhance clarity and depth, making your findings even more impactful.

**Do you want your identity to be public for this peer review?** For information about this choice, including consent withdrawal, please see our Privacy Policy

Reviewer #1: **Yes: ** Ramesh Ath

Reviewer #2: No

---

## [Author Response · Author response to Decision Letter 1]

1 Nov 2024

Dear Editor

Thank you very much for allowing us to revise the manuscript, where we have focused on improving the statistical results related to the data and methods sections, as well as making language and grammatical changes.

We hope that this manuscript will provide better content and quality for our readers.

Dear reviewers

Many thanks to the reviewers for their positive and constructive comments and suggestions on our manuscript. Both reviewers made good comments on the logic, data, models, methods, and conclusions of the article, which were helpful for us to improve our research.

We have carefully studied your comments and tried our best to revise our manuscript based on them. In the latest submitted manuscript�marked-up copy�, we have marked the deletions with red strikethroughs, the additions with blue colors, and the grammatical problems with green colors. Below is our response to the comments of the two reviewers, with the responses in blue.

Reviewer #1:

1. Abstract part needs revision, the result and conclusion part can be extended to provide better information to the readers.

We are very grateful for the reviewers' suggestions, and we have refined the writing of the abstract section and expanded the results and conclusions sections. Hopefully this will be a good solution to the problem raised by the reviewers.

2. The author’s mentioned the following statement, “However, Chinese scholars are divided on the specific path of integration.” Any specific reason; please make it clear for readers. Reference missing.

Thank you very much for the comment, you pointed out the missing references and the lack of specific elaboration in our paper, we decided to improve it. In order to better explain the different implementation options proposed by different scholars, and to explain why the Chinese government finally chose to integrate urban-rural residents' medical insurance first, we have changed the original text to the following:

At present, Chinese scholars have three representative paths for the integration of basic medical insurance, the first of which is to merge the New Rural Cooperative and the Urban Residents' Medical Insurance to form a basic medical insurance for residents, and then merge it with the basic medical insurance for urban workers, and finally to establish a nationwide national health insurance. The second is to make it mandatory for all people to enroll in urban and rural health insurance, and then integrate urban and rural health insurance when the urbanization rate reaches 70% to 80%. The third is to abolish the urban workers' health insurance, so that all citizens will participate in the national health insurance as unified residents, with the same level of individual contributions and government subsidies. However, the second path is limited by the differences between urban and rural areas, and it is difficult to realize the integration of the system in the short term. The third path can quickly achieve equal participation in the system, but the equalization of contributions fails to solve the problem of inequality caused by individual differences.

Because the urban and rural residents' health insurance systems are more homogeneous and closer in terms of funding levels, financial subsidies, and reimbursement ratios, they are easier to merge and can solve the problems of duplicated participation and duplicated subsidies. Therefore, Chinese scholars generally agree that the integration of urban and rural residents' health insurance should be realized first.

3. Authors mentioned, “Nationwide implementation marks a practical choice to pursue the integration path by starting with the integration of medical insurance for urban and rural residents”. The readers would have difficulty to extract what the authors try to say here.

We would like to thank the reviewer for the comments. The sentence “Nationwide implementation marks a practical choice to pursue the integration path by starting with the integration of medical insurance for urban and rural residents. The phrase “medical insurance for urban and rural residents” is indeed unclear and ambiguous. What we are trying to convey is that the Chinese government's implementation of Nationwide implementation of medical insurance for urban and rural residents in 2016 marks a practical choice to pursue the integration path by starting with the integration of medical insurance for urban and rural residents under the Three-Step Strategy. The new formulation has been revised in the text.

4. Authors mentioned the statement “existing studies ignore the hospital level” Any specific reason; please make it clear for readers. Reference missing.

We appreciate the reviewer's comment that the sentence ‘existing studies ignore the hospital level’ was not clear, and we have rephrased the sentence to better explain it to the readers and added literature on the effect of health insurance on healthcare behaviors. The revised text reads:

In addition, some of the studies have only examined the effect of health insurance on whether or not to seek medical care and the number of visits, and to a lesser extent, the effect of health insurance on a patient's choice of provider level.

5. Hypothesis of the present study missing.

Thank you for raising the medical, after checking, we did omit the hypotheses. In the revised text, we have added the relevant analysis and the hypotheses of this paper.

The hypothesis is that urban-rural health insurance coordination increases the utilization behavior of healthcare resources by releasing residents' demand for healthcare and triggering moral hazard, thus increasing residents' healthcare costs.

6. Research question and Aim of the study: the objective is not well phrased. They are not clear on what they are aiming to address. Clarity missing.

We are very grateful to the reviewers for their comments, and after checking, our article did not clearly state the research question and purpose. To solve this problem, we have added the following paragraph in the introduction:

Based on the above analyses, the purpose of this paper is to assess the policy performance of China's implementation of medical insurance for urban and rural residents, including exploring the impact of the urban-rural medical insurance integration on the utilization of healthcare resources and healthcare costs of Chinese residents and explaining it from the perspectives of both demand release and moral hazard.

7. The present study “organized and integrated data from 2013, 2015, and 2018”. It seems outdated data and predictions/implementation may not reach the objectives.

Many thanks for your comments, in the writing of the previous manuscript, although the data in the CHARLS database had been updated to 2020, we simply excluded the 2020 data due to the concern that the COVID-19 pandemic in 2020 would affect the data results. After the reviewers raised this issue, we reconsidered the use of the data, and our revised proposal was to use the data from 2013-2020 for the empirical analysis and exclude the data from 2020 in the robustness test section to explore whether the COVID-19 pandemic affected the data results of this study.

8. DID abbreviation must be mentioned in Introduction section and authors mentioned in methodology session.

Many thanks to your comments, we have rephrased the full name of the DID as well as the PSM model in the introduction section and added the names of relevant scholars in the methodology.

9. The authors used the following statement “There has also been a significant increase in the level of hospitals visited”. Any specific reason; please make it clear for readers.

Thank you very much for your advice, reviewer 2 made a similar suggestion that our explanation of the table was not clear enough. After checking, we really did not explain the sentence clearly. In response, we have added the relevant explanation in the text, and the refined formulation reads:

The results of the P-value test show that the sample's hospitalization visits, out-of-pocket expenses, and medical consumption significantly increase after the implementation of the health insurance integration. Meanwhile, compared with the base period 2013-2015, the proportion of the sample going to primary and intermediate hospitals in the experimental period 2018-2020 decreased, while the proportion of going to senior hospitals increased, indicating that the co-ordination improves the level of the residents' choice of medical institutions to visit.

10. Table 1. Result indicated age is statistically significant. Any specific reason; please make it clear for readers. The interpretation of Tables is not clear.

Thank you very much for the comments, age is indeed an important factor in health insurance selection and we appreciate you pointing out our problem. In addition, we also found that we did not explain the changes in variables other than the core variables, and to better address this issue, we have refreshed our interpretation of Table 1:

The results in Table 1 show that the age of the sample in the experimental period increases significantly compared to the base period, which may be due to the fact that senior patients are more inclined to purchase health insurance and the URRBMI policy improves the level of treatment and attracts more residents, especially senior residents, to purchase health insurance. The results in Table 1 also show that the proportion of the sample in the experimental period that is educated and married has significantly increased, which may be related to the fact that residents who are educated and married have higher health awareness and therefore purchase more urban and rural health insurance. Meanwhile, the health level of the sample in the experimental period increased significantly, which may be attributed to the health effect of the URRBMI.

Among the core variables, the probability of physical exercise, the proportion of non-smokers, and the proportion of those who participated in routine medical checkups increased in the sample in the experimental period, suggesting that the URRBMI may have increased the health awareness of the sample. At the same time, the probability of inpatient visits increased in the sample during the experimental period, with a significant increase in OOP inpatient costs and medical consumption., suggesting that the URRBMI may have increased residents' medical costs by increasing their healthcare resource utilization behaviors. Finally, the proportion of the sample going to primary and secondary hospitals decreased while the proportion going to senior hospitals increased in the experimental period, suggesting that the URRBMI may have led to a moral hazard problem by encouraging the residents to go to a higher level of healthcare institutions, which in turn may have led to an increase in the medical costs.

11. The authors did not mention anywhere details of “statistical methods were used in data and other analysis…” information throughout the manuscript. May it is a very important section for the study in terms of reader understands.

Many thanks for your comments, the methods of analyzing the descriptive statistics we have written below the tables in the form of comments, may not be obvious. To make it easier for the reader to understand, we have clarified the statistical methods of the data in the text.

We have redescribed the statistical methods of data analysis in the following sections:

(1) Inclusion and exclusion selection criteria for the sample were added to the Data Source and Study Sample section. The specific formulation is:

We collated and integrated the data for 2013, 2015, 2018, and 2020, and then excluded 6734 samples that were insured with basic urban workers' health insurance, 7167 samples that purchased other health insurance and private health insurance, and excluded samples that were missing key individual characteristic scalars such as age and gender to obtain a sample of 58,670 observations (S4 Fig.), with 25,670 in the experimental group and 33,000 in the control group.

2�The descriptive statistics section was supplemented with a methodology for between-group mean statistics. Table 1 calculates the changes in core variables and the significance of differences between groups in the sample before and after participation in URRBMI in 2013-2015 and 2018-2020, with means (scale) and t-tests for continuous variables; n (%) and chi-square tests for categorical variables.

3�Double difference model:

The double difference method was first applied to the field of economics by Ashenfelter (1978), and Chinese scholars Zhou and Chen (2005) introduced the DID model to China and applied it to the evaluation of public policies.

The DID model estimates the average treatment effect of a policy or intervention by comparing the difference between the treatment and control groups at two points in time before and after the policy or intervention. Assessing the policy effect under the traditional method is mainly done by setting a dummy variable for whether the policy occurs or not and then regressing it, in contrast, the model setup of the DID method is more scientific and can estimate the policy effect more accurately, and it can also avoid the problem of endogeneity to a certain extent.

4�Propensity score matching method

The core idea of the PSM method is to estimate the policy effect very effectively by calculating the propensity scores of the treatment group and the control group, and then performing kernel matching or other matching methods in order to find the individuals in the control group who are most similar to the treatment group, so that the two groups of data have similar characteristics in terms of variable observations, which can exclude the influence of other factors (Hirano et al., 2003) This study firstly uses the PSM method to match the individual characteristics of residents, controlling for the effect on the demand for healthcare because of their personal characteristics such as age, gender, income, and education, and then evaluated using the DID method, which aims to remove the effect of the release of healthcare demand on the increase in healthcare costs and to identify the phenomenon of induced demand in the integration.

12. The fixed-effects model will calculate only within the data variation, what about between data?

Many thanks for your comments, which were invaluable. Through reviewing the literature we learned that fixed-effects models can control the trend of individual changes over time and are more in line with real-world situations, while random-effects models, although controlling for changes between data, cannot identify all the differences between individuals, thus underestimating the standard errors (Gardiner & Luo�2010)

Meanwhile, most of the models related to health insurance policy evaluation by scholars such as Ren et al. (2022), Li et al. (2023), and Huang &Wu (2020) used fixed effects models. Therefore, this study used a fixed-effects double-difference model in the assessment concerning the integration of urban and rural residents' health insurance and residents’ health care costs. You raised this thought because we did not introduce the fixed-effects model in detail and did not distinguish the fixed-effects model from the random-effects model, to improve this problem, we rephrased the model part and added references.

13. Over all manuscript; flow, lack of interpretation and data analysis at tables including citation. For example: Ref: 40: “Rubin D B. Estimating causal effects of treatments in randomized and nonrandomized studies. Journal of educational Psychology.1974, 66(5): 688. And also Ref: 42-44”

We are very grateful to you for the comments, and to address this issue, we have provided more detailed descriptions of the corresponding positions.

(1) the explanations of Tables 1-Table 6 have been added.

(2) Explanation of reference 40. We have further supplemented the explanation of reference 40.

(3) Explanation of references 42-44. The authors did not mention anywhere details of ‘statistical methods were used in data and other analysis ...’ information throughout the manuscript. ‘ information throughout the manuscript’, our original manuscript di

---

## [Decision Letter · Decision Letter 1]

How did the Urban and Rural Resident Basic Medical Insurance integration Affect Medical Costs? ——Evidence from China

PONE-D-24-17636R1

Dear Dr. Su,

We’re pleased to inform you that your manuscript has been judged scientifically suitable for publication and will be formally accepted for publication once it meets all outstanding technical requirements.

Kind regards,

Patrick Goymer

Staff Editor

PLOS ONE

Additional Editor Comments (optional):

Reviewers' comments:

Reviewer's Responses to Questions

**Comments to the Author**

Reviewer #2: All comments have been addressed

2. Is the manuscript technically sound, and do the data support the conclusions?

Reviewer #2: Yes

3. Has the statistical analysis been performed appropriately and rigorously?

Reviewer #2: Yes

4. Have the authors made all data underlying the findings in their manuscript fully available?

Reviewer #2: Yes

5. Is the manuscript presented in an intelligible fashion and written in standard English?

Reviewer #2: Yes

Reviewer #2: I think reviewer address all the comments and suggestions for the improvement of quality. The revised version can be accepted for the publication

**Do you want your identity to be public for this peer review?** For information about this choice, including consent withdrawal, please see our Privacy Policy

Reviewer #2: **Yes: ** Dr Ghanshyam Pandey

---

## [Editor Report · Acceptance letter]

PONE-D-24-17636R1

PLOS ONE

Dear Dr. Su,

I'm pleased to inform you that your manuscript has been deemed suitable for publication in PLOS ONE. Congratulations! Your manuscript is now being handed over to our production team.

Kind regards,

on behalf of

Dr Patrick Goymer

Staff Editor

PLOS ONE